# Evaluation of nutritional supplements prescribed, its associated cost and patients knowledge, attitude and practice towards nutraceuticals: A hospital based cross-sectional study in Kavrepalanchok, Nepal

**Rabi Shrestha[1], Sweta Shrestha[2]\*, Badri K. C.[2‡], Sunil Shrestha[3‡]**

**1** Department of Pharmacy, Scheer Memorial Adventist Hospital, Banepa, Kavre, Nepal, **2** Department of Pharmacy, School of Sciences, Kathmandu University, Dhulikhel, Kavre, Nepal, **3** Department of Pharmaceutical and Health Service Research, Nepal Health Research and Innovation Foundation, Lalitpur, Nepal

☯ These authors contributed equally to this work.
‡ These authors also contributed equally to this work.
\* sweta.shrestha@ku.edu.np

## Abstract

### Background

There is substantial increment in nutraceutical consumption in Nepal, although the data on its efficacy and safety is scarce. The practices of nutraceutical supplements users in Nepal remain undocumented. Therefore, this study was conducted to study the prescription pattern, cost, knowledge, attitude and practice (KAP) of the patient towards nutraceutical.

### Methods

Descriptive cross-sectional study with stratified purposive sampling (n = 400) (patients from the out-patient departments of Scheer Memorial Adventist Hospital, Kavre, Nepal) was performed using a validated structured questionnaire assessing the socio-demographic characteristics, knowledge, attitude, practice of nutraceutical and total cost patients spent on nutraceutical alone. Pearson Chi-square test ($x^2$) was used to investigate the association between socio-demographic variables and patients' KAP (knowledge, attitude and practice) towards nutraceutical. One way ANOVA was performed to compare the cost of nutraceutical among the different outpatient departments.

### Results

More than 80% of patients were found to be consuming nutraceutical on their own. The mostly prescribed nutraceutical were vitamins (40.7%), minerals (23.7%), enzymes (21.1%), proteins (8.8%), probiotics (4.2%) and herbals (2.0%). With the most common reasons for consuming nutraceutical were to maintain good health (70.0%) and healthcare professionals (57.85%) were the most approached source of information for nutraceutical. Nearly half of the patients (46.5%) had an inadequate level of knowledge whereas more

**Data Availability Statement:** All relevant data are within the paper and its supporting information files.

**Funding:** The author(s) received no specific funding for this work.

**Competing interests:** The authors have declared that no competing interests exist.

than two-third (71.5%) showed a moderate positive attitude towards nutraceutical use. The average amount patients spent was NRs.575.78 [equivalent to USD 4.85] per prescription on nutraceutical alone. The maximum cost amounted to NRs 757.18 [equivalent to USD 6.43] in Orthopedics, and the minimum cost was NRs 399.03 [equivalent to USD 3.36] in Obstetrics and gynecology, respectively. There was a significant difference ($p$ <0.001) in cost of nutraceutical prescribed between the OPD clinics.

## Conclusion

The higher prevalence of inadequate knowledge despite moderate positive attitude towards nutraceutical among patients regarding some significant issues such as safety and interactions of nutraceutical consumption and its' substitution for meals reflects the need to develop an educational strategy to increase general public awareness on the rational use of nutraceutical.

## Introduction

Nutraceuticals is a term coined from the words "nutrition" and "pharmaceutical" which has been defined as "*food or part of a food that provides medical or health benefits including the prevention and/or treatment of a disease*" [1, 2]. The global data indicates the growing trend of functional food and nutraceutical market compared to the traditional food market [3]. This could be further supported by the Price waterhouse Coopers (PWC) analysis that has shown sharp growth in sales of vitamins and dietary supplements in global market by 6.3% (compound annual growth rate (CAGR) 2014–2018) and resulting the market size valued at USD 382.51 billion in 2019. It is expected that the market size further expands by 8.3% from 2016–2027 [4]. The global pharmaceutical manufacturing market size was valued at USD 324.42 billion in 2019 and is expected to grow at a CAGR of 13.74% from 2020 to 2027. Despite higher market size of a nutraceutical compared to pharmaceutical, there is lack in strict regulation of nutraceutical products. This could be evidenced by individuals easy access to nutraceuticals through supermarkets, pharmacies and internet [5]. The likelihood of in vivo clinical data to determine and assess nutraceutical safety and efficacy is relatively low and hence it could raise concern over the proven benefit from their usage [6].

The fastest-growing market for nutraceutical is South East Asia, with an estimated CAGR of 12% [7]. The higher use of nutraceutical is reflected by its emerging as a mainstream product and becoming a part of consumers' daily diet. The primary reasons for this dramatic shift can be attributed to rapid urbanization, increased prevalence of lifestyle diseases and people consciously taking preventive healthcare measures in form of dietary supplements [4]. Although a group of non-governmental organizations are advocating for including dietary supplements and nutraceuticals in the World Health Organization (WHO) model list of essential medicines to increase their access, WHO has not yet declared the inclusion of those products till date, from the ongoing consultation [8].

Meanwhile adverse event (AE) reporting for all the medicines is compulsory, only a very few countries have similar regulation for dietary supplements. In US, Congress passed the Dietary Supplement and Nonprescription Drug Consumer Protection Act in 2006. Between 2008 and 2011, the Food and Drug Administration (FDA) received over 4300 AE reports from the industry, but before enacting it used to be voluntary with many reports from other than the industry [9]. The efficacy of fish oils (e.g., cod liver oil) in the diet has been demonstrated in

several clinical trials, animal feeding experiments and *in vitro* models that mimic cartilage destruction in arthritic disease. Also, there is some evidence for beneficial effects of other nutraceuticals, such as green tea, herbal extracts, chondroitin sulphate and glucosamine [10]. However, many dietary supplements lack scientific evidence for or against the use, and yet they continue to be widely ingested throughout the world [11].

In context to Nepal, Dietary Supplement Guideline has been published by the Ministry of Agriculture and Livestock Development, regulating the nutraceuticals. However, unwarranted prescribing of nutraceuticals by physicians for financial benefits from the manufacturers has been reported continuously. Department of drug administration (DDA) has been attempting to stop such practices, putting the patients at a high economic burden and possible health hazard. Risky behaviors like the purchase of unauthorized herbal formulations and nutraceuticals from street vendors are prevalent in Nepal, which must be brought to a complete halt [12].

There are numerous nutraceutical products that have been claimed to offer a cure for many diseases, thus misleading chronic disease patients to use these products although scientific evidence backing such claims are lacking [13]. The claims of an energy booster, weight loss, enhanced appearance and other miraculous benefits for different illnesses may have prompted many consumers to fall for these unscientific assertions [14]. Lack of stringent regulations to benchmark nutraceutical use reflects the need to explore the current prescription practice and consumption of nutraceuticals among patients. Furthermore, it becomes essential to investigate the patients' knowledge, attitude and practice of nutraceutical to customize the educational programs to ensure the appropriate use of these supplements. With this background, the current study was conducted considering the paucity of data regarding prescribing of nutraceuticals and the lack of information of patients' knowledge, attitude, and practice of nutraceuticals in context of Nepal.

## Materials and methods

### Study design, study site, population characteristics and ethical approval

A descriptive cross-sectional study was conducted at Scheer Memorial Adventist Hospital (SMAH), Kavrepalanchok, Nepal. SMAH has 150-bed facilities and is located near Kathmandu, the capital city of Nepal, and provides a full range of out-patient and in-patient services. Ethical approval was obtained before data collection from the Scheer Memorial Adventist Hospital (SMAH) Ethical Review Committee.

The study population consisted of patients who visited the out-patient department (OPD) of SMAH (medicine, surgery, orthopedic, and obstetrics and gynecology) during the study period of three months from June 2019 to August 2019. Patients prescribed with nutraceuticals visiting the OPD of SMAH (medicine, surgery, orthopedic, and obstetrics and gynecology) and thus agreed to provide written consent to participate in the study were included. The excluded criteria included patients visiting emergency departments and mentally disabled patients.

Privacy and confidentiality were maintained by not disclosing the name of the participants and ensuring them, that collected information was used only for the study purpose. The parents or guardians were interviewed, and they provided informed consent in representation of the minors as a part of the study. The interviewer administered the questionnaire to each participant and any explanation required was provided to the respondents.

### Sampling method and technique

The sample size was calculated using the Raosoft online calculator [15]. Raosoft online calculator is designed specifically for population surveys to calculate the sample size and determine

how many responses are needed to meet the desired confidence level with the margin of error (usually 5%) [15]. According to the hospital census, the total population of patients visiting OPD is 22500 (with an average 250 per day) in a three-month duration. Therefore, to achieve a confidence level of 95% and a 5% margin of error and 50% response distribution, a minimum sample size of 378 was required. Stratified purposive sampling was performed. The SMAH has mainly 5 outpatient clinics; medicine, orthopedic, gynecology, surgery and pediatrics. Pediatrics were excluded, and data collection was done in remaining four departments. Total of 400 patients and 100 from each department were taken to distribute sample in strata properly.

Estimation of the minimum number of respondents to be included can be obtained by using the formula given below,

$$n = N.Z\alpha/2^2.P(1-P)/[(N-1)e^2 + Z\alpha/2^2.P(1-P)]$$

Where,
n = minimum sample size of the study subject
N = total number patient visited in three-month duration as per census
Z = standard normal distribution curve /value at $\alpha/2$ for the 95% confidence interval (1.96)
P = proportion of the population with KAP of nutraceutical among respondents (50%)
e = the margin of error (0.05).

## Data collection tool and technique

An interviewer-administered questionnaire (See S1 File) was designed after reviewing the previous similar studies with some modifications [16–18]. The questionnaire was reviewed and subjected to a validation process. The questionnaire was tested for readability and comprehensibility among 38 patients (10% of 378) visiting SMAH. Colleagues from the pharmacy department did face validation of the questionnaire and the content validation of the data collection tool was conducted by discussing the questionnaire with content experts of pharmacy practice, physicians, nutrition expert and statistician. Cronbach's alpha value was calculated as a measure of the internal consistency of the questionnaire which was found to be 0.783. The outcomes of the pre-testing were not included in the final data analysis.

The final version of the data collection tool comprised of four sections. Section 1 consisted of 7 questions investigating the demographic and related information of patients: age, gender, religion, degree/education, marital status and occupation. Section 2 comprised of 8 items to evaluate the practice of nutraceutical use among patients. Section 3 comprised of 3 items aimed to evaluate patients' knowledge of nutraceutical and section 4 included 14 items designed to evaluate the attitude of patients towards nutraceutical.

**Scoring system.** To assess the respondents' knowledge on nutraceutical, each correct answer was coded as 'yes' and scored as '1', and an incorrect answer was coded as 'no' and scored as '0'. Any "I do not know" response was also scored as '0'. The cumulative and mean scores were calculated. Based on the mean score obtained, participant's knowledge was categorized as "Adequate" (0.75–1), "moderately adequate" (0.5–0.749) and "inadequate" (<0.5). To assess respondents' attitude, a five-point Likert scale was used that ranges from; "Strongly agree" i.e., '1' to "Strongly disagree" i.e., '5'. Based on the mean score, the respondents who scored above the mean score were defined as having a "positive attitude" (4–5), "moderate positive attitude" (3–3.99) and those who scored below the mean score were defined as having a "negative attitude"(<3).

**Cost calculation.** The cost for the individual category of nutraceutical and other medications were noted. The total cost that the patient spent on nutraceutical compared to the total cost of prescribed drugs was calculated using the formula below:

Average nutraceutical cost % on prescription = (Total cost of nutraceutical/Total prescription cost) $^*100\%$

## Data management and analysis

Data was entered and analyzed using Statistical Package for Social Sciences (SPSS) version 20 (SPSS Inc., Chicago, IL, USA) and p <0.05 was considered as statistically significant in all the analyses. Descriptive analysis was performed using frequencies and percentages. Data was presented in the form of text, figures, and tables. Pearson Chi-square test ($x^2$) for independence was used to determine the association among socio-demographic variables and patients' knowledge, attitude and practice (KAP) towards nutraceutical. One-way Analysis of variance (ANOVA) test was used to compare the cost of nutraceutical among different departments.

## Results

### Socio-demographic characteristics

Table 1 shows the socio-demographic characteristics of the patients visiting the out-patient department of SMAH Kavrepalanchok, Nepal. The majority (40%, n = 160) of the participants

**Table 1. Socio-demographic characteristics of the respondents (N = 400).**

| Characteristics | Frequency (n) | Percentage (%) |
|---|---|---|
| **Age (in years)** | | |
| 15–29 | 122 | 30.50 |
| 30–44 | 160 | 40 |
| 45–59 | 99 | 24.75 |
| > 60 | 19 | 24.75 |
| **Gender** | | |
| Male | 189 | 47.25 |
| Female | 211 | 52.75 |
| **Ethnicity** | | |
| Brahmin | 50 | 12.50 |
| Chhetri | 85 | 21.25 |
| Newar | 134 | 33.50 |
| Others | 131 | 32.75 |
| **Religion** | | |
| Hindu | 305 | 76.25 |
| Buddhist | 67 | 16.75 |
| Christian | 21 | 5.25 |
| Others | 7 | 1.75 |
| **Degree/Education** | | |
| Illiterate | 28 | 7 |
| Able to read and write | 42 | 10.50 |
| Primary Level | 35 | 8.75 |
| Secondary Level | 120 | 30 |
| Higher secondary and above | 175 | 8.75 |
| **Occupation** | | |
| Housewife | 105 | 26.25 |
| Service | 112 | 28 |
| Business | 86 | 21.50 |
| Others | 97 | 24.25 |

were of the age group 30–44. Female participants were higher (52.75%, n = 211), and the majority were Newar (33.50%, n = 134) and followed Hindu religion (76.25%, n = 305). Most participants had an education level of higher secondary and above (43.75%, n = 175) and were service holders (28%, n = 112).

### Prescribing pattern of nutraceuticals

More than three quarters of patients (n = 325, 81.2%) were already using nutraceutical. Frequency of prescribing of different nutraceutical was observed in the following departments: medicine OPD, surgery OPD, orthopedic OPD and Obstetrics and gynecology OPD. Vitamins, Minerals, Enzymes, Proteins, Probiotics and Herbals in decreasing frequency orders were the most frequently prescribed nutraceuticals in the OPD departments (Table 2).

An average number of drugs and nutraceuticals prescribed per prescription were 3.42 and 1.52, respectively.

### Gender wise distribution of knowledge and practice on nutraceutical use

Majority of the participants (46.5%, n = 186) had inadequate knowledge followed by moderate adequate knowledge (36.30%, n = 145) whereas only few (17.3%, n = 69) had adequate knowledge on nutraceutical. Most of the participants (90.2%, n = 361), especially the females (90.5%, n = 191) claimed to know about nutraceutical with no significant difference between the male and female ($p$ = 0.847). More than one-third of participants (38%, n = 152) believed that nutraceutical is safe to consume and majority 36.2% (n = 145) opined that nutraceutical interacts with drug, food or drinks with no significant difference between male and female participants ($p$ = 0.812) and ($p$ = 0.293) respectively.

There was no significant difference in the consumption rate of nutraceutical between male and female ($p$ = 0.37). Majority of both male and female respondents (87.5%, n = 350) believed that counselling is essential before consuming nutraceuticals ($p$ = 0.664), but a significantly higher proportion of female (90%, n = 190) consulted for professional medical help when taking nutraceuticals ($p$ = 0.004). Health professionals were the most approached source of information for nutraceuticals by majority of the participants, (57.85%, n = 231) followed by multimedia (17.5%), books (12%), friends (8.2%) and others (4.5%). However, no significant difference between male and female partakers ($p$ = 0.173) was observed in this regard. About half of the respondents (49.2%, n = 197) believed that the information available on nutraceuticals was adequate, with a significantly higher proportion of female undertaking this belief ($p$ = 0.005).

More than two-third (72.5%, n = 290) of the respondents bought nutraceuticals with the prescription, and there was no significant difference in the self-medication practice between male and female ($p$ = 0.26). Majority of them (58%, n = 232) bought nutraceuticals from the hospital pharmacy, followed by community pharmacy (34.8%), others (4.8%) and department

**Table 2. Percentage of nutraceuticals prescribed.**

| S. N. | Nutraceuticals | Frequency (N = 540) | Percentage (%) |
|---|---|---|---|
| 1 | Vitamins | 220 | 40.7 |
| 2 | Minerals | 128 | 23.7 |
| 3 | Enzymes | 114 | 21.1 |
| 4 | Proteins | 48 | 8.8 |
| 5 | Probiotics | 23 | 4.2 |
| 6 | Herbal | 11 | 2 |

**Table 3. Gender wise distribution of knowledge and practice question (Chi-square test).**

| Question (Knowledge) | Options | Total participants (%) | Male (%) | Female (%) | p-Value |
|---|---|---|---|---|---|
| Do you know what nutraceutical are? | Yes | 361 (90.2%) | 170(89.9%) | 191(90.5%) | 0.847 |
| | No | 39 (9.8%) | 19 (10.1%) | 20 (9.5%) | |
| Are uses of nutraceuticals always safe? | Yes | 152 (38%) | 70 (37.0%) | 82 (38.9%) | 0.812 |
| | No | 127 (31.8%) | 63 (33.3%) | 64 (30.3%) | |
| | Don't Know | 121 (30.2%) | 56 (29.6%) | 65 (30.8%) | |
| Can drug/food/drinks interact with nutraceutical? | Yes | 145 (36.2%) | 63 (33.3%) | 82 (38.9%) | 0.293 |
| | No | 85 (21.2%) | 38 (20.1%) | 47 (22.3%) | |
| | Don't Know | 170 (42.5%) | 88 (46.6%) | 82 (38.9%) | |
| **Question (Practice)** | | | | | |
| Do you use any nutraceutical? | Yes | 325 (81.2%) | 147 (77.8%) | 178 (84.4%) | 0.092 |
| | No | 75 (18.8%) | 42 (22.2%) | 33 (15.6%) | |
| How often do you consume nutraceutical? | Occasionally | 222 (55.5%) | 111 (58.7%) | 111 (52.6%) | 0.37 |
| | 3 to 5 times a week | 58 (14.5%) | 23 (12.2%) | 35 (16.6%) | |
| | Daily | 82 (20.5%) | 35 (18.5%) | 47 (22.3%) | |
| | Not sure | 38 (9.5%) | 20 (10.6%) | 18 (8.5%) | |
| Is counseling on nutraceutical important? | Yes | 350 (87.5%) | 168 (88.9%) | 182 (86.3%) | 0.664 |
| | No | 22 (5.5%) | 10 (5.3%) | 12 (5.7%) | |
| | Don't Know | 28 (7.0%) | 11 (5.8%) | 17 (8.1%) | |
| Do you consult medical personnel for nutraceutical? | Yes | 341 (85.2%) | 151 (79.9%) | 190 (90.0%) | 0.004 |
| | No | 59 (14.8%) | 38 (20.1%) | 21 (10.0%) | |
| What is the source of information? | Health professionals | 231 (57.8%) | 100 (52.9%) | 131 (62.1%) | 0.173 |
| | Friends | 33 (8.2%) | 15 (7.9%) | 18 (8.5%) | |
| | Multimedia | 70 (17.5%) | 39 (20.6%) | 31 (14.7%) | |
| | Books | 48 (12.0%) | 28 (14.8%) | 20 (9.5%) | |
| | Others | 18 (4.5%) | 11 (5.2%) | 7 (3.7%) | |
| Is information available are adequate? | Yes | 197 (49.2%) | 77 (40.7%) | 120 (56.9%) | 0.005 |
| | No | 115 (28.7%) | 62 (32.8%) | 53 (25.1%) | |
| | Don't Know | 88 (22.0%) | 50 (26.5%) | 38 (18.0%) | |
| How do you buy nutraceutical? | With prescription | 290 (72.5%) | 132 (69.8%) | 158 (74.9%) | 0.26 |
| | Without prescription | 110 (27.5%) | 57 (30.2%) | 53 (25.1%) | |
| From where did you buy nutraceutical? | Hospital pharmacy | 232 (58.0%) | 106 (56.1%) | 126 (59.7%) | 0.502 |
| | Community Pharmacy | 139 (34.8%) | 67 (35.4%) | 72 (34.1%) | |
| | Department store | 10 (2.5%) | 7 (3.7%) | 3 (1.4%) | |
| | Others | 19 (4.8%) | 9 (4.8%) | 10 (4.7%) | |
| **Summary of Knowledge Level** | **F (n = 400)** | | **Percentage (%)** | | |
| Level of Knowledge (mean score) | | | | | |
| Inadequate Knowledge (<0.5) | 186 | | 46.5 | | |
| Moderate Adequate Knowledge (0.5–0.749) | 145 | | 36.3 | | |
| Adequate Knowledge (0.75–1) | 69 | | 17.3 | | |

store (2.5%) with no significant difference in place of buying between male and female participants ($p$ = 0.502) (Table 3).

## Patients' attitude on nutraceutical use

Table 4 presents the patients attitude on nutraceutical use. Majority of the participants (69%, n = 276), believed that nutraceutical serve as energy booster, enhance physical appearance (74.3%, n = 297), is necessary for all ages (77.3%, n = 309), is harmless (65.5%, n = 262) and

**Table 4. Percentage distribution of respondents' attitude towards nutraceutical.**

| S. NO. | Attitude variables | Percentage distribution of respondents' attitude towards nutraceutical | | | | |
|---|---|---|---|---|---|---|
| | | Strongly Agree n (%) | Agree n (%) | Neutral n (%) | Disagree n (%) | Strongly Disagree n (%) |
| 1 | Nutraceuticals are needed if a person feels tired and rundown. | 90 (22.5) | 186 (46.5) | 47 (11.8) | 52 (13.0) | 25 (6.3) |
| 2 | Nutraceutical make one feel better physically. | 77 (19.3) | 228 (57.0) | 50 (12.5) | 28 (7.0) | 17 (4.3) |
| 3 | Nutraceuticals usually improve a person's appearance. | 75 (18.8) | 222 (55.5) | 44 (11.0) | 34 (8.5) | 25 (6.3) |
| 4 | Body fat can be lost by taking certain type of nutraceuticals. | 24 (6.0) | 132 (33.0) | 125 (31.3) | 88 (22.0) | 31 (7.8) |
| 5 | One can skip meals and just take nutraceuticals. | 7 (1.8) | 32 (8.0) | 45 (11.3) | 210 (52.5) | 106 (26.5) |
| 6 | The nutrients supplied by food need to be supplemented. | 45 (11.3) | 232 (58.0) | 56 (14.0) | 46 (11.5) | 21 (5.3) |
| 7 | Nutraceuticals is necessary for all ages. | 107 (26.8) | 202 (50.5) | 27 (6.8) | 28 (7.0) | 36 (9.0) |
| 8 | Nutraceuticals is generally harmless. | 28 (7) | 234 (58.5) | 58 (14.5) | 52 (13.0) | 28 (7.0) |
| 9 | Regular use of supplements prevents chronic diseases | 54 (13.5) | 188 (47.0) | 90 (22.5) | 42 (10.5) | 26 (6.5) |
| 10 | Nutraceuticals can prevent cancers. | 27 (6.8) | 114 (28.5) | 136 (34.0) | 79 (19.8) | 44 (11.0) |
| 11 | Health professional should promote use of supplements | 48 (12.0) | 221 (55.3) | 71 (17.8) | 44 (11.0) | 16 (4.0) |
| 12 | Nutraceuticals should be sold only on prescription of a registered medical practitioner. | 77 (19.3) | 207 (51.7) | 34 (8.5) | 59 (14.8) | 23 (5.8) |
| 13 | Manufacture and sale of nutraceuticals should be monitored by a regulatory body. | 123 (30.8) | 206 (51.5) | 38 (9.5) | 17 (4.3) | 16 (4.0) |
| 14 | Use of Nutraceutical are just waste of money. | 4 (1.0) | 12 (3.0) | 31 (7.8) | 194 (48.5) | 159 (39.8) |

can prevent chronic diseases (60.5%, n = 242) or even cancers (35.3%, n = 141). Most of them believed that healthcare professionals should promote the use of nutraceutical (67.3%, n = 269) which should as well be looked upon by some regulatory body (82.3%, n = 329) and it should be categorized as a prescription-only medicine (71%, n = 284). Nutraceuticals use accounts for unnecessary expenditure was disagreed by a maximal number of participants (88.3%, n = 353). The statement that nutrition received from food is inadequate and requires supplementation from nutraceutical was accorded by 69.3% (n = 277), and it aids in losing body fat was consented by 39% (n = 156). Some also believed that nutraceutical could be a substitute for meals (9.8%, n = 39). Majority (21.3%, n = 85) had a negative attitude, 71.5% (n = 286) had a moderately positive attitude, whereas only 7.2% (n = 29) of participants possessed positive attitude regarding the use of nutraceutical.

## Association between socio-demographic variables and level of knowledge and attitude

The overall data on associations are presented in Table 5. There was no association between the socio-demographic variables and the patients' knowledge level on nutraceutical ($p > 0.05$). A significant association was found between the age of patients ($p < 0.001$) and their attitude. Association between age group 30–44 years and moderate positive attitude and positive attitude was found to be highly significant ($p < 0.001$). Similarly, there was an association between age group 45–49 years and positive attitude ($p = 0.001$). Furthermore, there was significant association between the age group above 60 years and negative attitude, moderate positive attitude, and positive attitude ($p = 0.002$, $p < 0.001$, $p = 0.003$, respectively). No association was there with other demographic variables regarding attitude on nutraceutical.

## Reasons for nutraceutical use

Majority of the patients (70%, n = 280) thought that nutraceutical had been prescribed for maintaining good health (12%, n = 48) reportedly perceived its use for enhancing appearance,

**Table 5. Association between the level of knowledge and attitude and their selected socio-demographic variables.**

| Variables | Total Participants 400 (%) | Level of knowledge | | | | | Level of attitude | | | | |
|---|---|---|---|---|---|---|---|---|---|---|---|
| | | Inadequate Knowledge | Moderate Adequate Knowledge | Adequate Knowledge | Chi-square value | Sig | Negative Attitude | Moderate Positive Attitude | Positive Attitude | Chi square value | Sig |
| **Age (in years)** | | | | | | | | | | | |
| 15–29 | 122 (30.5%) | 52 (42.6%) | 51 (41.8%) | 19 (15.6%) | 6.337 | 0.386 | 25 (20.5) | 87 (71.3) | 10 (8.2) | 24.377 | **0.00** |
| 30–44 | 160 (40.0%) | 82 (51.2%) | 51 (31.9%) | 27 (16.9%) | | | 34 (21.2) | 123 (76.9) | 3 (1.9) | | |
| 45–59 | 99 (24.8%) | 46 (46.5%) | 33 (33.3%) | 20 (20.2%) | | | 19 (19.2) | 69 (69.7) | 11 (11.1) | | |
| Above 60 | 19 (4.8%) | 6 (31.6%) | 10 (52.6%) | 3 (15.8%) | | | 7 (36.8) | 7 (36.8) | 5 (26.3) | | |
| **Gender** | | | | | | | | | | | |
| Male | 189 (47.2%) | 91 (48.1%) | 68 (36.0%) | 30 (15.9%) | 0.61 | 0.737 | 45 (23.8) | 128 (67.7) | 16 (8.5) | 2.549 | 0.28 |
| Female | 211 (52.8%) | 95 (45%) | 77 (36.5%) | 39 (18.5%) | | | 40 (19.0) | 158 (74.9) | 13 (6.2) | | |
| **Ethnicity** | | | | | | | | | | | |
| Brahmin | 50 (12.5%) | 20 (40%) | 20 (40%) | 10 (20%) | 6.772 | 0.342 | 14 (28.0) | 34 (68.0) | 2 (4.0) | 7.109 | 0.311 |
| Chhetri | 85 (21.2%) | 42 (49.4%) | 34 (40%) | 9 (10.6%) | | | 19 (22.4) | 56 (65.9) | 10 (11.8) | | |
| Newar | 134 (33.5%) | 68 (50.7%) | 45 (33.6%) | 21 (15.7%) | | | 30 (22.4) | 97 (72.4) | 7 (5.2) | | |
| Others | 131 (32.8%) | 56 (42.7%) | 46 (35.1%) | 29 (22.1%) | | | 22 (16.8) | 99 (75.6) | 10 (7.6) | | |
| **Religion** | | | | | | | | | | | |
| Hindu | 305 (76.2%) | 138 (45.2%) | 112 (36.7%) | 55 (18%) | 10.79 | 0.095 | 62 (20.3) | 219 (71.8) | 24 (7.9) | 2.159 | 0.90 |
| Buddhist | 67 (16.8%) | 36 (53.7%0 | 24 (35.8%) | 7 (10.4%) | | | 17 (25.4) | 46 (68.7) | 4 (6.0) | | |
| Christian | 21 (5.2%) | 10 (47.6%) | 8 (38.1%) | 3 (14.3%) | | | 5 (23.8) | 15 (71.4) | 1 (4.8) | | |
| Others | 7 (1.8%) | 2 (28.6%) | 1 (14.3%) | 4 (57.1%) | | | 1 (14.3) | 6 (85.7) | 0 | | |
| **Education** | | | | | | | | | | | |
| Illiterate | 30 (7.5%) | 17 (56.7%0 | 10 (33.3%) | 3 (10%) | 4.205 | 0.838 | 8 (26.7) | 17 (56.7) | 5 (16.7) | 11.77 | 0.162 |
| Just Read and write | 42 (10.5%) | 21 (50%) | 12 (28.6%) | 9 (21.4%) | | | 6 (14.3) | 30 (71.4) | 6 (14.3) | | |
| Primary Level | 34 (8.5%) | 17 (50%) | 11 (32.4%) | 6 (17.6%) | | | 8 (23.5) | 23 (67.6) | 3 (8.8) | | |
| Secondary Level | 119 (29.8%) | 51 (42.9%) | 45 (37.8%) | 23 (19.3%) | | | 23 (19.3) | 89 (74.8) | 7 (5.9) | | |
| Higher secondary and above | 175 (43.8%) | 80 (45.7%) | 67 (38.3%) | 28 (16%) | | | 40 (22.9) | 127 (72.6) | 8 (4.6) | | |
| **Marital Status** | | | | | | | | | | | |
| Married | 352 (88%) | 167 (47.4%) | 125 (35.5%) | 60 (17%) | 6.473 | 0.167 | 75 (21.3) | 252 (71.6) | 25 (7.1) | 0.646 | 0.958 |
| Unmarried | 45 (11.2%) | 19 (42.2%) | 19 (42.2%) | 7 (15.6%) | | | 9 (20.0) | 32 (71.1) | 4 (8.9) | | |
| Divorced | 3 (0.8%) | 0 | 1 (33.3%) | 2 (66.7%) | | | 1 (33.3) | 2 (66.7) | 0 | | |
| **Occupation** | | | | | | | | | | | |
| Housewife | 105 (26.2%) | 51 (48.6%) | 38 (36.2%) | 16 (15.2%) | 4.621 | 0.593 | 22 (21.0) | 75 (71.4) | 8 (7.6) | 1.617 | 0.951 |
| Service | 112 (28.0%0 | 49 (43.8%) | 46 (41.1%) | 17 (15.2%) | | | 23 (20.5) | 82 (73.2) | 7 (6.2) | | |
| Business | 86 (21.5%) | 41 (47.7%) | 25 (29.1%) | 20 (23.3%) | | | 21 (24.4) | 60 (69.8) | 5 (5.8) | | |
| Others | 97 (24.2%) | 45 (46.4%) | 36 (37.1%) | 16 (16.5%) | | | 19 (19.6) | 69 (71.1) | 9 (9/3) | | |

(4.5%, n = 18) believed weight loss is the reason for its prescription and very few (3.75%, n = 15) thought that nutraceutical was prescribed with no reason (Table 6).

## Cost comparison of prescribed nutraceuticals

The present study showed that the average amount that a patient spent on nutraceutical was NRs. 575.78 [equivalent to USD 4.85] per prescription, the maximum accounted to NRs. 757.18 [equivalent to USD 6.38] in Orthopedics and minimum ranged to NRs. 399.03

**Table 6. Patient's perception on reason for nutraceutical use.**

| Reason | Total 400 (%) |
|---|---|
| Maintain good health | 280 (70) |
| Treatment of disease | 157 (39.25%) |
| Prevent disease | 142 (35.5%) |
| Ensure adequate nutrition | 125 (31.25%) |
| Meet increased energy needs | 97 (24.25%) |
| Enhance appearance | 48 (12%) |
| Weight Loss | 18 (4.5%) |
| No specific reason | 15 (3.75%) |

[equivalent to USD 3.36] in obstetrics and gynecology OPD. A significant difference was found in the cost of nutraceuticals among the various OPD ($p <$0.001). Based on a post hoc test (LSD) analysis, there was a significant difference in the cost of nutraceutical prescribed among the various OPD clinics except in medicine–obstetrics and gynecology OPD ($p = 0.067$). Table 7 shows details.

## Discussion

The current study aimed to quantify the frequency of concomitant use of nutraceutical, an important area of ongoing clinical concern along with assessing patients' preferences related to information sources consulted and patients' KAP regarding nutraceutical. Our data shows that there is a huge gap between the knowledge and practices among the patients for nutraceutical use in Nepal, suggesting the need for educational intervention to increase public awareness on the rational use of nutraceutical.

The present study showed that majority of the patients (81.2%) were already consuming nutraceutical. This demonstrates that use of nutraceutical by patients has become a common practice for a myriad of reasons in Nepal. For instance, the preference to the use of these nutraceutical could be explained by individual's perception that nutraceutical aids in treating their ailments and lack of nutraceutical is accountable for various disorders [19]. Further to these, physicians perception towards nutraceutical as having beneficial effect has also resulted the possible rise in the prescription of nutraceutical in out-patient departments [20] as observed in our study. Besides, individuals' increased consciousness towards their health has led to an escalated demand for nutritious products as a health booster. Nutraceutical thus has gained popularity in the general population as an essential element of a regular balanced diet [17, 21]. Vitamins were the most widely prescribed nutraceutical in our study, followed by minerals, enzymes, and herbal. This finding is in line of agreement with the study conducted in India among the health science students [22]. A similar relatively high consumption (43%, n = 105) of vitamin-mineral supplements has been observed among the university students in Malaysia [20].

**Table 7. Comparison of cost of nutraceutical prescribed department wise (One way ANOVA).**

| OPD | N | Mean ± SE | 95% Confidence Interval for Mean | | P -Value |
|---|---|---|---|---|---|
| | | | Lower Bound | Upper Bound | |
| Medicine | 100 | 511.95 ± 44.75 | 423.15 | 600.75 | |
| Surgical | 100 | 634.99 ± 44.60 | 546.48 | 723.49 | |
| Orthopedic | 100 | 757.18 ± 48.30 | 661.32 | **853.03** | **<0.001** |
| Obstetrics and Gynecology | 100 | 399.03 ± 35.05 | 329.47 | 468.58 | |
| Total | 400 | 575.78 ± 22.66 | 531.23 | 620.33 | |

Despite the higher use of nutraceutical, our study showed that very less proportion of patients (17.3%) were aware of the term nutraceutical. Our finding showed a huge difference with the previous study conducted by Navyashree et al., which reported a higher proportion (76%, n = 38) were aware of the term nutraceutical [23]. A total of 31.8% of participants believed that nutraceutical is not always devoid of side effects which is lesser than that reported by the author (72%, n = 36). Majority of the participants (46.5%, n = 186) had inadequate knowledge of nutraceutical. This difference may be accounted to the fact that the author had conducted the study among medical practitioners in a tertiary care hospital, unlike our study, which assessed the KAP of patients visiting the hospital. An overestimation of nutraceutical safety among people should be a matter of concern as reported by this study. Regulation of dietary supplements by the FDA is relatively liberal in terms of safety requirements. However, the consumption of dietary supplements is not without risk. Photosensitivity and neurotoxicity at higher doses of pyridoxine, toxicity resulting from increased consumption of fat-soluble vitamins, an association of congenital abnormalities with increased prevalence of vitamin A consumption during pregnancy have been reported. Hence contemplation of potential interactions and adverse effects of dietary supplements is essential to avoid harmful medical sequel. It is equally important for the consumers and the physicians to refer to evidence-based literature or credible resources before initiating such supplements. Since dietary supplements have become a common, physicians and pharmacists should inquire well about supplement intake history to avoid any possible drug supplement interactions [24].

Majority of the participants (81.2%, n = 325) in our study were using some kind of nutraceutical, showing a huge differences in data as presented by Alhoumoud et al. (37%, n = 74) among the non-medical students in one of the universities of United Arab Emirates (UAE) [16]. Most of the participants (85.2%, n = 341) sought professional medical help before taking nutraceutical, which is higher than that reported by the author (44%, n = 88) [16]. Furthermore, (87.5%, n = 350) of our participants believed that counselling is essential before consuming nutraceutical. This reflects the trust put upon the healthcare professionals by the patients and their expectation of receiving credible information from the healthcare professionals on nutraceutical. The fact that maximum patients purchase nutraceutical from nearby hospital pharmacy (58%, n = 232) following physicians' prescription (72.5%, n = 290) supports the statement mentioned above.

Regarding the attitude towards nutraceuticals, majority of participants perceived that nutraceuticals prevent chronic diseases (60.5%, n = 242) or even cancer (35.3%, n = 141) in contrary to the findings of Gosavi et al., where none of the patients believed that nutraceuticals had been prescribed to prevent chronic illnesses [17]. Despite the prominent use of nutraceuticals, a sheer lack of clinical evidence to reinforce the health claim of chronic disease or cancer prevention has been reported [18, 24]. The majority felt the need for some regulatory body to survey the manufacture and sale of nutraceuticals and voted for its' categorization under prescription-only products. The participants were also in favor of healthcare professionals promoting the use of nutraceutical. This result is consistent with Navyashree et al., where the medical practitioners stated the need to regulate nutraceuticals and were against its availability as over the counter (OTC) products [23]. A small proportion of the patients, 9.8%,, also believed that nutraceutical could be a complete substitute for meals which is slightly lower to the data obtained by Cruz et al. where 17.5% possessed such perceptions [14]. Such beliefs among people might lead to behaviors like skipping of meals and overuse of nutraceuticals. Hence, we, the present authors, suggest the attention of healthcare professionals must be drawn to such areas where patients need to be counselled appropriately.

Our study showed that majority of the patients (70%), mainly perceived nutraceutical as drugs which help to maintain good health, 39.25% thought the use can aid in the treatment of

their illnesses 31.25% ensure adequate nutrition and 35.5% prevent diseases. Similar findings were reported by Teoh SL et al. in which maintenance of good health, preventing future illnesses along with perceived health benefits of nutraceuticals especially in the current situation where consumption of fruits and vegetables loaded with pesticides pose a significant challenge in sustaining health condition of the general public were identified as the main factors affecting the consumer's decision to consume nutraceutical [18].

The reasons for nutraceutical use outlined in our research are also consistent with the findings of a few other studies conducted among university students in Malaysia and Saudi Arabia [22, 25, 26]. Nevertheless, reasons like improving mental performance, body function and general well-being recorded by Abdullah and Waquar [27] were not mentioned by our participants. Despite the potential benefits of nutraceutical, the public needs to understand that a balanced diet and healthy lifestyle are essential facets of good health, and nutraceutical alone cannot be a substitute. Furthermore, consumers must be aware that nutritional supplements should not be looked upon as replacements of medical therapy to cure or treat a disease condition, and equally valid because supplements are not devoid of side effects and interactions with other medications [16].

Healthcare professionals, internet, television, followed by books were reported to be the most common sources of information on nutraceutical. This observations was not surprising as this aligns with the previous studies finding [22, 26]. Healthcare professionals, especially physicians and pharmacists, remain in a highly trusted position to offer drug or disease-related credible information and therefore we suggest that there should be updated knowledge on the use of nutraceutical among healthcare professionals. Besides, in this era of virtually limitless information, healthcare professionals should offer guidance to the users on identifying and obtaining credible sources of information on nutraceutical. Physicians, pharmacists and books can provide more trustworthy and evidence-based information to nutraceutical users compared to television advertisement [16].

Our study showing that maximum of NRs. 757.18 [equivalent to USD 6.38] was spent in Orthopedics and a minimum of NRs. 399.03 [equivalent to USD 3.36] spent in Obstetrics and gynecology department by the patients is in the range of previous study suggesting the similar expenditure in respective department [17]. Accordingly, we also found that 88.3% (n = 353) of the participants did not regard expenditure on nutraceuticals as a waste of money. Instead, our observation showed that many of the participants believed nutraceuticals serving as a supplement source of nutrition that cannot be adequately received from food. Nutraceuticals are gaining a widespread market share globally, with China and India taking giant leaps as the fast-moving markets [12] and thus Nepal cannot remain untouched by this influential global growth rate. Although, the price of nutraceuticals has been identified as a determining factor to its purchase and consumption, its use has also been anticipated by many as a way of preventing chronic illnesses which may cost them a more significant deal of money [18].

With the observations in context to nutraceutical use in Nepal, the present authors suggest health benefits claimed by manufacturers of functional foods or food ingredients with health outcomes needs to be backed by scientific evidence and significant scientific agreement. Given the market appeal and claims made by nutraceutical, sufficient legislations to safeguard the health of the population are urgently warranted in Nepal.

## Conclusion

There is a relatively high prevalence of nutraceutical consumption among patients due to their high perception that it is necessary to maintain good health and ensure adequate nutrition. The high prevalence may be due to the information suggested by the healthcare professionals,

most of whom consume it following a physician's prescription. Our study shows that most of the patients possessed a moderately positive attitude but nearly one half of the study population had inadequate knowledge regarding nutraceutical. With these observations, we suggest awareness of nutraceutical use, and information should be integrated into everyday practice to ensure the proper use of nutraceutical related products. There is a continuing need for nutraceutical consumers to be educated to determine the appropriate use of these supplements.

## Strength and limitations

To the best of our knowledge, this is the first study in Nepal to date reporting the nutraceutical use among patients. Similarly, this is the first study in the country that has attempted to address the gap regarding KAP of patients on nutraceuticals. This study offers the baseline data to other researchers willing to perform large scale research on the relevant topic.

However, our study has some limitations. The study was a cross-sectional study, and therefore, it gave only a snapshot of participants KAP relating to nutraceutical. It was carried out only in a particular hospital, and the result obtained from this cannot be generalized at the national level. This study assessed the KAP of only those who were either consuming or were prescribed with nutraceutical, which can lead to participant bias and/or response bias for assessing KAP of nutraceutical. Furthermore, the influence of predictor variables and their corelation for KAP was not included in the present study.

## Recommendation

Studies assessing the KAP of healthcare professionals regarding nutraceuticals in Nepal is an important area requiring exploration. Nutraceutical use among the general public needs to be studied on a larger scale. There is a call for interventional studies that can improve the KAP of patients and consumers on nutraceutical.

## Supporting information

**S1 File. Questionnaire English version.** KAP questionnaire.
(DOCX)

**S1 Data. Data in Excel.**
(XLSX)

**S2 Data. Data in SPSS.**
(SAV)

## Acknowledgments

The author (s) would like to thank Associate Professor Dr Rajani Shakya, Head of Department of Pharmacy, Kathmandu University and Mr Tara Jung Gurung, PhD for their co-operation and extreme support and guidance. The author (s) would like to thank Scheer Memorial Adventist Hospital, Banepa, Kavre, Nepal, for permission to conduct the study. The author(s) would like to acknowledge all the study participants.

## Author Contributions

**Conceptualization:** Rabi Shrestha, Sweta Shrestha.

**Data curation:** Rabi Shrestha, Sunil Shrestha.

**Formal analysis:** Rabi Shrestha, Sunil Shrestha.

**Investigation:** Rabi Shrestha.

**Supervision:** Sweta Shrestha, Badri K. C.

**Writing – original draft:** Rabi Shrestha.

**Writing – review & editing:** Sweta Shrestha, Badri K. C., Sunil Shrestha.

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
