## [Decision Letter · Decision Letter 0]

22 Dec 2020

PONE-D-20-36161

Prescription pattern and costs of nutraceuticals visiting out-patient department in Nepal and their knowledge, attitude, practice towards nutraceuticals: A cross-sectional study

PLOS ONE

Dear Dr. shrestha,

Thank you for submitting your manuscript to PLOS ONE. After careful consideration, we feel that it has merit but does not fully meet PLOS ONE’s publication criteria as it currently stands. Therefore, we invite you to submit a revised version of the manuscript that addresses the points raised during the review process.

We look forward to receiving your revised manuscript.

Kind regards,

Jenny Wilkinson, PhD

Academic Editor

PLOS ONE

Journal Requirements:

2.) We suggest you thoroughly copyedit your manuscript for language usage, spelling, and grammar. If you do not know anyone who can help you do this, you may wish to consider employing a professional scientific editing service.  

3.) You indicated that you had ethical approval for your study.

In your Methods section, please ensure you have also stated whether you obtained consent from parents or guardians of the minors included in the study or whether the research ethics committee or IRB specifically waived the need for their consent.

4.) Please include captions for your Supporting Information files at the end of your manuscript, and update any in-text citations to match accordingly. Please see our Supporting Information guidelines for more information: http://journals.plos.org/plosone/s/supporting-information

Additional Editor Comments:

Thank you for your submission. Reviewers have provided detailed comments which are provided for your consideration. Both reviewers have also noted that the manuscript need editing for English language and you may wish to seek assistance from a professional editing service for this task.

Reviewers' comments:

Reviewer's Responses to Questions

**Comments to the Author**

1. Is the manuscript technically sound, and do the data support the conclusions?

Reviewer #1: Partly

Reviewer #2: No

2. Has the statistical analysis been performed appropriately and rigorously? 

Reviewer #1: I Don't Know

Reviewer #2: I Don't Know

3. Have the authors made all data underlying the findings in their manuscript fully available?

Reviewer #1: Yes

Reviewer #2: No

4. Is the manuscript presented in an intelligible fashion and written in standard English?

Reviewer #1: No

Reviewer #2: No

5. Review Comments to the Author

Reviewer #1: Title: Title statement (“……visiting OPD in Nepal and”) seems strange it should rather be “..Hospital OPD in Nepal” . As categorizations/ operational definition of nutraceutical wasn't clear (see below comments in method section ), "More Precisely, I suggest the title something like “Evaluation of Nutritional supplements prescribed, its associated Cost and Patients knowledge, attitude and practice towards Nutraceuticals: A hospital based cross-sectional study in Kabrepalanchok, Nepal”

Abstract :

In methods, please mention the statistical test used during data analysis.

Introduction:

Citation of the original source (instead of reference provided) is recommended particularly in paragraph 2. Author are advised not to provide the assumed hypothetical or contradictory statements regarding study intention and outcomes of study as laid down in paragraph 4 and 5 of the introduction section beside the rationale of the study needs be properly stated. The flow and smoothness in and during transition of paragraph need to be improved.

I also wish to also see a literature body related to advantages/ disadvantages associated with chaotic use of nutraceuticals, factors influencing consumer’s/prescribers attitude regarding nutraceuticals, marketing and regulatory facts regarding the sale and distribution of Nutraceutical particularly in content to Nepal all in one to two paragraph within the acceptable word count limit.

Methods:

Inclusion criteria: you have mentioned “those who were prescribed nutraceuticals” will be enrolled in study. Though the inclusion is appropriate for evaluating of prescribing pattern. However, this can strongly introduce participant bias and/or response bias for assessing KAP of nutraceutical.

Exclusion criteria: what about the geriatric and/or mentally retarded patients?

sampling techniques: Author have mentioned “stratified random sampling” in abstract whereas stratified purposive sampling is stated in methods section. Author is advice to reconcile the contradict section. please mention how much sample size is taken from each departments with justification.

Data collection tools and techniques:

Author is advised to cite the references article reviewed or undertaken while designing the tools/questionnaire of the study. Please state the mean score value for adequate, moderately adequate and inadequate knowledge, similarly for positive, moderately positive and negative attitude.

Prescribing pattern categories of nutraceutical which you have laid down is not clear. Is there any appropriate references followed to classify nutraceutical? In my understanding. There are marginal differences between the nutraceutical (which are generally understood for food derived dietary supplements) and pharmaceutical supplements (which actually contains pharmaceutically active ingredients, properly dose regulated and quality controlled e.g specific active vitamins supplement, folic acid, calcium and iron tablets). While both are nutritional health supplements, differentiation is important for better clarity (as pharmaceutical single or multiple doses form supplements are widely used /and are important in medical therapy while nutraceutical are often comes a dietary supplements and are not necessarily regulated and doesn’t require prescription, particularly in country like Nepal). Therefore, proper characterization of the types of nutraceutical needs to be done as per study aim else the readers may misinterpret the findings.

Data analysis and Results:

Prescribing pattern of Nutraceuticals:

You have mention in inclusion criteria as “Patients who were prescribed nutraceutical” are enrolled in study. The first line reads 80 % of patients are using/prescribed nutraceuticals? please clarify.

Table 2. Kindly find comments in methods section regarding categorization of nutraceuticals. I don’t think as per your study aim, there is any significance of tabulating the nutritional supplements categories as per the use by departments.

Tables could be made more concise and reader friendly , for e.g presenting the gender wise distribution of knowledge, attitude related table (table 3 and 8) in a single table, and similarly association related table (table 4 and 6) in another table, while illustrating the significant results in text.

Data analysis remarks: you have used chi square test to determine the relationship between sociodemographic groups and KA level ( many of those consisting with 4/5*3 consistency table). Analysis with chi square to determine relationship in such cases may not be appropriate, and in many instances requires post hoc test for chi square for identifying where the significant association lies (if any) .

Alternatively, distributing relevant sociodemographic variables (such as age, gender, literacy level, Marital status, employment status etc) wisely into binary group and then getting it compared with frequency score (using chi square) or with the mean/median score of Knowledge and attitude (e.g using t test, Anova, Mann-whitney, KW test for means which ever suitable) may be more appropriate.

Besides, determining the influence of predictor variables like age, literacy status, employment status etc and/or their co-relation for KAP will further strengthen the findings?

Table 7. Since it’s the patient’s perceived response to nutraceutical been prescribed, I don’t see any rationale for presenting it in a distributive form with respect to OPD clinics.

Table 9. Finding of Post hoc test for ANOVA can be explained in text and need not necessarily be presented in table.

Discussion:

Discussion needs refinements.

Strength and implication can be provided after the conclusion instead of first paragraph.

Some of the important background information are being highlighted in discussion (can be rearrange to Introduction section)

The transition of paragraph needs to be properly arranged as per the order of finding of the result. Author are advised to compared their findings to more of the relevant/similar research literature of other developed and developing countries including that of South-Asian countries in terms of nutraceutical use and patients KAP.

Conclusion: Conclusion should be based only as per the finding of your study. Additional suggestion and recommendation could be made separately in recommendation section.

General comments: The article requires improvements in English, possible proof reading by native English speaker or professional expert/ services is recommended.

Reviewer #2: Overall impression, the language and formatting were poorly done. The major concern about the paper is the methodological aspect, especially the sampling population and the content of the questionnaire. The methods despite not clearly described, do not seem to me to have reached a scientific enough level.

Some other specific comments:

Abstract

1) Not convinced with the implication of use of the research that has been described in the abstract's conclusion.

2) Please use a more appropriate language for "Vitamins were prescribed maximum in medicine ".

Introduction

1) Line 82-87 describes about the use of nutraceuticals for malnutrition using WHO's statistics. I am not agreeable to the purpose of the use of nutraceuticals in your context. I believe the participants here were taking nutraceuticals for "dietary supplementation" instead of "malnutrition". Malnutrition is normally for those in critical environment.

Method

1) Line 119-120 "The total population of a patient visiting OPD was 22500 (with an average 250 per day) in three-month

duration according to hospital census." should be in results instead of methods or after the descriptions of sampling method.

2) Line 124 - 127 - "A questionnaire was designed after reviewing the previous similar studies with some

modifications. Structured questions were used for the collection of data, and the prescriptions and

bills were scrutinized" - I am not convinced about the appropriateness of the content of the developed questionnaires.

3) There were no referred versions to be seen. No experts to examine the content of questionnaire?

3) Line 128 - participants were randomly selected? How? It was not described

4) Line 149-150 - How the calculation of "the average percentage of costs spent on nutraceutical

150 alone compared to the total prescribed drugs were measured following OPD." is performed, it is not clearly described.

Results

1) 201 - "Do you know what nutraceutical are?" Does not seem to be accurately testing the content of the "knowledge" of participants. "Nutraceuticals" itself is a very jargonish term even researcher of healthcare providers may not be aware of.

6. PLOS authors have the option to publish the peer review history of their article (what does this mean?). If published, this will include your full peer review and any attached files.

Reviewer #1: No

Reviewer #2: No

---

## [Author Response · Author response to Decision Letter 0]

18 Feb 2021

Dear Editor

We have submitted all the changes and addressed all the comments of the reviewers in the tracked changed version of manuscript and the response sheet.

---

## [Editor Report · Decision Letter 1]

3 Mar 2021

PONE-D-20-36161R1

Evaluation of nutritional supplements prescribed, its associated cost and patients knowledge, attitude and practice towards Nutraceuticals: A hospital based cross-sectional study in Kavrepalanchowk, Nepal

PLOS ONE

Dear Dr. shrestha,

Thank you for submitting your manuscript to PLOS ONE. After careful consideration, we feel that it has merit but does not fully meet PLOS ONE’s publication criteria as it currently stands. Therefore, we invite you to submit a revised version of the manuscript that addresses the points raised during the review process.

We look forward to receiving your revised manuscript.

Kind regards,

Jenny Wilkinson, PhD

Academic Editor

PLOS ONE

Journal Requirements:

Additional Editor Comments (if provided):

Thank you for your responses; these have largely addressed the reviewer comments. There are however still several grammatical and other language errors and I encourage you to seek assistance from a native English writer or professional editorial service. I also ask that you use only a single track change author as multiple authors result in many different coloured track changes which makes it quite difficult to read.

---

## [Author Response · Author response to Decision Letter 1]

16 Apr 2021

16th April, 2021

Response to reviewers

Dear Editor, 

Thank you very much for providing us the comments on improving the manuscript level as per the Plos One standard. We believe that the revised version of the manuscript has been improved as a result of the reviewers’ comments and feedback and the re-revised version has been successful in meeting the Plos One standard, particularly improving the English level (as per the additional editor comments) and the editor comments on cross-checking the references to respond the retracted article in proper way (if any).

Please find the revised manuscript “Evaluation of nutritional supplements prescribed, its associated cost and patients knowledge, attitude and practice towards Nutraceuticals: A hospital based cross-sectional study in Kavrepalanchowk, Nepal”. 

As per the suggestion from editor, we have used the fluent English writer to have a look at our manuscript and thus we believe that the quality of the manuscript has been improved. We also have used single level of track changes mode-on for not adding difficulty to editor to assess the changes made. At some places, the manuscript has been edited for brevity.

Best wishes

Sweta Shrestha

---

## [Editor Report · Decision Letter 2]

4 May 2021

PONE-D-20-36161R2

Evaluation of nutritional supplements prescribed, its associated cost and patients knowledge, attitude and practice towards Nutraceuticals: A hospital based cross-sectional study in Kavrepalanchowk, Nepal

PLOS ONE

Dear Dr. shrestha,

Thank you for submitting your manuscript to PLOS ONE. After careful consideration, we feel that it has merit but does not fully meet PLOS ONE’s publication criteria as it currently stands. Therefore, we invite you to submit a revised version of the manuscript that addresses the points raised during the review process.

We look forward to receiving your revised manuscript.

Kind regards,

Jenny Wilkinson, PhD

Academic Editor

PLOS ONE

Journal Requirements:

Additional Editor Comments (if provided):

Thank you for your revisions, while the content related issues have been addressed there are still many grammatical and language issues. For example, the sentence “Many schools and universities started switching from traditional classroom teaching to virtual education methods” is repeated at the top of page 3, ‘et al’ is incorrectly shown with full stops after both et and al and author initials included in in-text citations, etc. I strong advise seeking assistance from a native English writer experience in academic writing or professional editing service.

---

## [Author Response · Author response to Decision Letter 2]

7 May 2021

Date: 7th May 2021

Jenny Wilkinson, PhD 

Academic Editor 

PLOS ONE

Subject: Response to Editor and submission of the revised version of the manuscript

Dear editor, 

Thank you very much for providing us with the comments on improving the manuscript level as per the Plos One standard. We have revised manuscript titled “Evaluation of nutritional supplements prescribed, its associated cost and patients knowledge, attitude and practice towards Nutraceuticals: A hospital based cross-sectional study in Kavrepalanchowk, Nepal”, Manuscript ID: PONE-D-20-36161R2 and submitting you the revised manuscript in both track changes and clean version.

We have mentioned a point-to-point response below to the comments by the editor.

Yours Sincerely,

Sweta Shrestha on behalf of the study team.

Comments:

Journal Requirements: Please review your reference list to ensure that it is complete and correct. If you have cited papers that have been retracted, please include the rationale for doing so in the manuscript text, or remove these references and replace them with relevant current references. Any changes to the reference list should be mentioned in the rebuttal letter that accompanies your revised manuscript. If you need to cite a retracted article, indicate the article’s retracted status in the References list and also include a citation and full reference for the retraction notice.

Response: We believe that the revised version of the manuscript has been improved because of the reviewers’ comments and feedback, and the re-revised version has been successful in meeting the Plos One standard, particularly improving the English level (as per the additional editor comments) and the editor comments on cross-checking the references to respond the retracted article properly (if any).

Additional Editor Comments (if provided):

Thank you for your revisions, while the content related issues have been addressed there are still many grammatical and language issues. For example, the sentence “Many schools and universities started switching from traditional classroom teaching to virtual education methods” is repeated at the top of page 3, ‘et al’ is incorrectly shown with full stops after both et and al and author initials included in in-text citations, etc. I strong advise seeking assistance from a native English writer experience in academic writing or professional editing service.

Response: The sentence “Many schools and universities started switching from traditional classroom teaching to virtual education methods” you mentioned is not mentioned in our manuscript. We believe that there is some confusion. Please kindly verify it once. As per the suggestion from the editor, we have used a fluent English writer to look at our manuscript, and thus we believe that the quality of the manuscript has been improved. We have also used the single level of track changes mode to make it easier for the editor to assess the changes made. At some places, the manuscript has been edited for brevity.

---

## [Editor Report · Decision Letter 3]

18 May 2021

Evaluation of nutritional supplements prescribed, its associated cost and patients knowledge, attitude and practice towards Nutraceuticals: A hospital based cross-sectional study in Kavrepalanchowk, Nepal

PONE-D-20-36161R3

Dear Dr. shrestha,

We’re pleased to inform you that your manuscript has been judged scientifically suitable for publication and will be formally accepted for publication once it meets all outstanding technical requirements.

Kind regards,

Jenny Wilkinson, PhD

Academic Editor

PLOS ONE
---

## [Editor Report · Acceptance letter]

24 May 2021

PONE-D-20-36161R3 

Evaluation of nutritional supplements prescribed, its associated cost and patients knowledge, attitude and practice towards Nutraceuticals: A hospital based cross-sectional study in Kavrepalanchowk, Nepal 

Dear Dr. Shrestha:

I'm pleased to inform you that your manuscript has been deemed suitable for publication in PLOS ONE. Congratulations! Your manuscript is now with our production department. 

Kind regards, 

on behalf of

Dr Jenny Wilkinson 

Academic Editor

PLOS ONE